# The Ret receptor regulates sensory neuron dendrite growth and integrin mediated adhesion

Peter Soba[1,2,4]*, Chun Han[2,3,4], Yi Zheng[2,4], Daniel Perea[5], Irene Miguel-Aliaga[5], Lily Yeh Jan[2,4], Yuh Nung Jan[2,4]*

[1]Center for Molecular Neurobiology, University Medical Center Hamburg-Eppendorf (UKE), University of Hamburg, Hamburg, Germany; [2]Department of Physiology, Howard Hughes Medical Institute, University of California, San Francisco, San Francisco, United States; [3]Weill Institute for Cell and Molecular Biology, Cornell University, Ithaca, United States; [4]Department of Biochemistry and Biophysics, University of California, San Francisco, San Francisco, United States; [5]Gut Signalling and Metabolism Group, MRC Clinical Sciences Centre, Imperial College London, London, United Kingdom

**Abstract** Neurons develop highly stereotyped receptive fields by coordinated growth of their dendrites. Although cell surface cues play a major role in this process, few dendrite specific signals have been identified to date. We conducted an in vivo RNAi screen in *Drosophila* class IV dendritic arborization (C4da) neurons and identified the conserved Ret receptor, known to play a role in axon guidance, as an important regulator of dendrite development. The loss of *Ret* results in severe dendrite defects due to loss of extracellular matrix adhesion, thus impairing growth within a 2D plane. We provide evidence that *Ret* interacts with integrins to regulate dendrite adhesion via *rac1*. In addition, Ret is required for dendrite stability and normal F-actin distribution suggesting it has an essential role in dendrite maintenance. We propose novel functions for Ret as a regulator in dendrite patterning and adhesion distinct from its role in axon guidance.

*For correspondence: peter.soba@zmnh.uni-hamburg.de (PS); yuhnung.jan@ucsf.edu (YNJ)

**Competing interests:** The authors declare that no competing interests exist.

## Introduction

Accurate functional connectivity and sensory perception require proper development of the neuronal dendritic field, which ultimately determines the (sensory) input a specific neuron can receive and detect. Thus, coordinated dendrite growth and patterning is important for establishing the often complex, but highly stereotyped organization of receptive fields. Two of the organizing principles in dendrite development are self-avoidance and tiling (*Grueber and Sagasti, 2010*; *Jan and Jan, 2010*; *Zipursky and Sanes, 2010*). While self-avoidance describes the phenomenon of recognition and repulsion of isoneuronal dendritic branches, tiling refers to the complete yet non-redundant coverage of a receptive field by neighboring neurons of the same type. Both phenomena have been described in different systems across species including the mouse, zebrafish, medicinal leech, *Caenorhabditis elegans*, and *Drosophila melanogaster*.

Dendritic patterning by self-avoidance, tiling, and other mechanisms is thought to be mediated by cell surface receptors and cell adhesion molecules (CAMs), which play a pivotal role in integrating environmental and cellular cues into appropriate growth and adhesion responses. Many such receptors, prominently Robo (*Spitzweck et al., 2010*) and Ephrin receptors (*Egea and Klein, 2007*), have well understood roles in axon guidance. Although some of these axonal cues including Robo/Slit play a role in dendrite development as well (*Dimitrova et al., 2008*; *Gibson et al., 2014*), dendritic

**eLife digest** There are hundreds of types of neurons, but all of them are variations on the same basic theme. Each neuron consists of a cell body, which contains the nucleus, and various structures that stick out from the cell body. These include a large number of short protrusions called dendrites, and a long thin cable-like structure called the axon. The dendrites receive incoming signals from the environment or neighboring neurons and transmit these signals to the cell body, which then relays them along the axon and on to the dendrites of the next neuron.

As the brain develops, newly formed dendrites recognize and repel other dendrites belonging to the same neuron, thereby spreading themselves out to occupy a larger volume. This patterning process is called self-avoidance. At the same time, in order to repel each other, the dendrites must encounter each other in the first place, which means that they need to grow on a common substrate or surface.

Soba et al. have now identified one of the proteins responsible for the self-avoidance process by studying the growth of dendrites on neurons in living fruit fly larvae. When the gene for a protein called the Ret receptor was deleted or inhibited, the dendrites that grew were shrunken and disorganized.

High-resolution microscopy revealed that the dendrites were usually anchored to a scaffolding structure called the extracellular matrix, which ensured that they could only grow in two dimensions. However, when the gene for the Ret receptor did not work properly, the dendrites detached from this matrix and grew in three dimensions instead. Further experiments revealed that this detachment occurred because the Ret receptor was no longer interacting with a group of structural proteins called integrins.

The Ret receptor plays a role in human disease and has previously been implicated in axon growth, but this is the first evidence to suggest that it also has a role in the patterning of dendrites. Given that Ret is present in vertebrates and has changed little over time, it is likely that this protein also helps to shape communication within the extensive networks of neurons that support complex cognitive functions in mammals.

surface receptors and their functions are not fully characterized to date. Recent efforts have yielded some progress in this area. *Down's syndrome cell adhesion molecule* (*Dscam*) has been shown to regulate dendrite self-avoidance in *Drosophila* (*Hughes et al., 2007*; *Matthews et al., 2007*; *Soba et al., 2007*) and mouse (*Fuerst et al., 2008*, *2009*). More recently, studies on protocadherins have revealed that they play an important role in dendrite self-avoidance in mammals (*Lefebvre et al., 2012*). In *C. elegans*, sax-7/L1-CAM and menorin (mnr-1) form a defined pattern in the surrounding hypodermal tissue to guide PVD sensory neuron dendrite growth via the neuronal receptor dma-1 (*Dong et al., 2013*; *Salzberg et al., 2013*). However, given the complexity and stereotypy of dendritic arbors within individual neuronal subtypes, it is important to search for additional signals for directing dendrite growth.

The *Drosophila* peripheral nervous system (PNS) has served as an excellent model which has helped to elucidate several molecular mechanisms regulating dendrite development (*Grueber and Sagasti, 2010*). The larval PNS contains segmentally repeated dendritic arborization (da) neurons which have been classified as class I–IV according to their increasing dendritic complexity (*Grueber et al., 2002*). All da neuron classes feature highly stereotyped sensory dendrite projections. Moreover, all da neurons exhibit self-avoidance behavior allowing them to develop their individual receptive fields without overlap. It has been demonstrated that all da neuron classes require *Dscam* for dendrite self-avoidance (*Hughes et al., 2007*; *Matthews et al., 2007*; *Soba et al., 2007*). In addition, the atypical cadherin *flamingo* (*Matsubara et al., 2011*) and immunoglobulin super family (IgSF) member *turtle* (*Long et al., 2009*) might play a more restricted role in C4da neuron self-avoidance. *Netrin* and its receptor *frazzled* have also been shown to act in parallel to *Dscam* in class III da neurons ensuring their proper dendritic field size and location by providing an attractive growth cue which is counterbalanced by self-avoidance (*Matthews and Grueber, 2011*). For tiling, no surface receptor has been identified to date. However, the conserved *hippo and tricornered* kinases, and more recently the *torc2* complex, have been implicated in C4da neuron tiling, as the loss of function of these genes results in iso- and heteroneuronal crossing of dendrites (*Emoto et al., 2004*, *2006*; *Koike-Kumagai et al., 2009*).

Recent work has further shown that dendrite substrate adhesion plays an essential role in patterning. Da neuron dendrites are normally confined to a 2D space through interaction with the epithelial cell layer and the extracellular matrix (ECM) on the basal side of the epidermis (*Yamamoto et al., 2006*; *Han et al., 2012*; *Kim et al., 2012*). 2D growth of da neuron dendrites requires integrins, as loss of the α-integrin *mew* (*multiple edomatous wing*) or ß-integrin *mys* (*myospheroid*) results in dendrites being freed from the 2D confinement due to detachment from the ECM. Thus, they can avoid dendrites by growing into the epidermis leading to 3D crossing of iso- and hetero-neuronal branches (*Han et al., 2012*; *Kim et al., 2012*). Integrins are therefore essential to ensure repulsion-mediated self-avoidance and tiling mechanisms, which restrict growth of dendrites competing for the same territory (*Han et al., 2012*; *Kim et al., 2012*). How integrins are recruited to dendrite adhesion sites and whether they cooperate with other cell surface receptors is unknown.

To identify novel receptors required for generating complex, stereotypical dendritic fields, we performed an in vivo RNAi screen for cell surface molecules in C4da neurons. We identified the *Drosophila* homolog of *Ret* (<u>re</u>arranged during <u>t</u>ransfection) as a patterning receptor of C4da dendrites. Loss of *Ret* function in C4da neurons severely affects dendrite coverage, dynamics, growth, and adhesion. In particular, dendrite stability and 2D growth are impaired resulting in reduced dendritic field coverage and abnormal 3D dendrite crossing, respectively. These defects can be completely rescued by Ret expression in C4da neurons. We further show that Ret interaction with integrins is needed to mediate C4da dendrite-ECM adhesion, but not dendrite growth. Our data suggest that *Ret* together with integrins acts through the small GTPase *rac1*, which is required for dendrite adhesion and 2D growth of C4da neuron dendrites as well. We thus describe a novel role for the Ret receptor in dendrite development and adhesion by direct receptor crosstalk with integrins and its downstream signals.

## Results

### Expression analysis and RNAi-mediated knockdown of Ret reveal its role in C4da neuron dendrite development

To identify cell surface receptors mediating dendrite development of C4da neurons we used an in vivo RNAi screening approach. We focused on functionally defined classes of proteins containing conserved extracellular domains, including IgSFs and receptor tyrosine kinases (RTKs). To this end we expressed available RNAi transgenes specifically in the embryonic/larval PNS (*21-7-Gal4*) together with a C4da neuron specific reporter line (*ppk*-CD4-tdTomato, *Han et al., 2012*). We also utilized a *UAS-Dcr2* transgene to enhance RNAi efficiency (*Dietzl et al., 2007*). Using this approach, we screened approximately 400 RNAi lines targeting IgSFs and RTKs and found that knockdown of Ret with two independent lines led to strong dendrite defects in C4da neurons (*Figure 1A*). Knockdown of Ret resulted in abnormal C4da dendrite patterning with crossing of dendritic branches and incomplete coverage of their receptive field. We did not observe defects in other classes of da neurons (*Figure 1—figure supplement 1* and data not shown) suggesting that Ret plays a specific role in C4da neuron dendrite morphogenesis.

*Drosophila Ret* encodes a highly conserved receptor tyrosine kinase (RTK) with neuronal expression (*Sugaya et al., 1994*; *Hahn and Bishop, 2001*; *Kallijärvi et al., 2012*). In order to validate our RNAi results, we tested the expression of Ret in C4da neurons. A Gal4 enhancer trap line inserted within the 5′-region of the *Ret* gene showed C4da neuron expression when driving a GFP reporter (*UAS-CD8-GFP*, *Figure 1B*). We next performed immunohistochemical analysis of larval filet preparations using Ret specific antibodies. We first used a commercial phospho-Ret antibody which showed a mostly somatic granular signal only in C4da neurons (*Figure 1—figure supplement 2A*). We tested its specificity with a mutant allele containing a P-element insertion within the 3′UTR region of the Ret gene (*Ret$^{C168}$*), which in combination with a deficiency covering the *Ret* locus (*Df(Bsc$^{312}$)*) led to a strong reduction of the Ret immunoreactivity in C4da neurons (*Figure 1—figure supplement 2A–C*). To validate and analyze Ret expression in C4da neurons in more detail, we raised a Ret specific antibody against its intracellular domain (see 'Materials and methods'). Interestingly, Ret was detected throughout the dendritic arbor showing almost continuous staining along major dendrites (*Figure 1C,C'*) and a more granular pattern in terminal dendrites (*Figure 1C,C''*). We detected granular Ret signal even in distal terminal dendrites suggesting that is localizing throughout the C4da dendritic arbor (*Figure 1D*). In addition, the strongest Ret puncta consistently localized to the basal side of dendrites facing the ECM (*Figure 1C',C''*). In *Ret* mutant animals, only residual

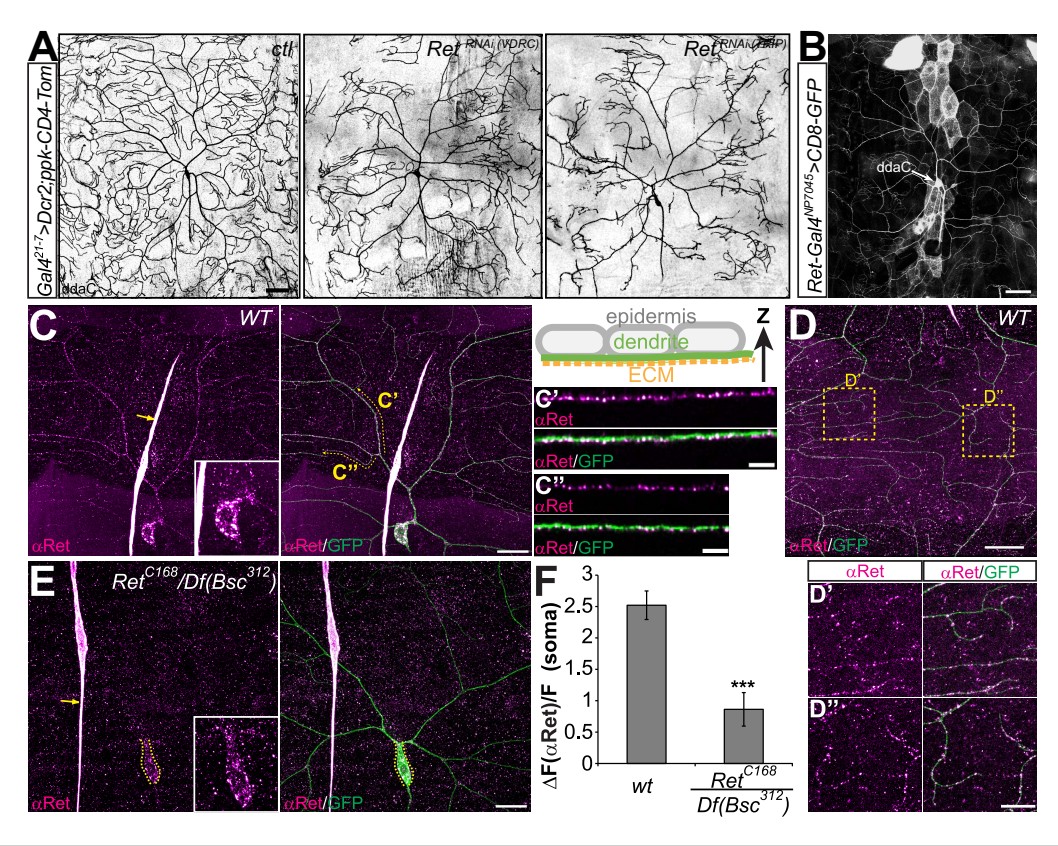

**Figure 1**. In vivo RNAi knockdown of *Ret* causes C4da neuron dendrite pattering defects. (**A**) *Ret^RNAi* transgenes together with *Dcr2* were driven by *Gal4^21–7* and C4da neuron morphology was visualized with a specific fluorescent reporter (*ppk-CD4-tdTomato*). Confocal live images of control animals (ctl) show wildtype dendrite morphology, while expression of either of two independent *Ret-RNAi* transgenes led to severely disorganized dendrites with incomplete receptive field coverage. (**B**) A Gal4 insertion in the *Ret* genomic locus drives CD8-GFP expression in C4da neurons indicating the presence of Ret. Scale bar 50 µm. (**C–E**) Immunohistochemical analysis of Ret expression in third instar larvae of wildtype (**C** and **D**) and *Ret* deficient animals (**E**). Overlays with GFP expressing C4da neurons (*ppk-Gal4 > CD4-tdGFP*) show that specific anti-Ret signal could be detected throughout the C4da dendritic arbor (**C**). Resliced portions (in Z direction) of primary (**C'**) and terminal (**C''**) C4da dendrites show that Ret strongly labels the basal side of dendrites facing the ECM as shown in the schematic drawing. (**D**) Ret was also present in distal terminal dendrites of the dorsal field of C4da neurons (**D'** and **D''**, scale bar 20 µm, 10 µm for insets). Residual Ret signal was detected in the C4da neuron soma in *Ret* mutant animals but not in dendrites (**E** inset). Arrows indicate non-specific antibody signal present in wildtype and *Ret* mutant samples. Scale bar 20 µm. (**F**) Quantitative analysis of Ret immunoreactivity in C4da somata of wildtype and *Ret* mutant samples showing the signal over background (ΔF/F, mean ± SD, n = 5, p < 0.001, Student's two-tailed *t*-test).

The following figure supplements are available for figure 1:

**Figure supplement 1**. Class I da neurons do not display obvious morphological defects in Ret mutant animals.

**Figure supplement 2**. *phospho*-Ret levels are reduced in Ret mutant C4da neurons.

immunoreactivity could be detected in C4da neuron somata but not in dendrites (*Figure 1E,E',F*) suggesting that the allelic combination used is hypomorphic. Together, these results reveal expression and subtype specific functions of Ret in C4da neurons.

## Loss of Ret function results in dendrite patterning and 3D crossing defects

We next wanted to corroborate the Ret-RNAi induced dendrite phenotype of C4da neurons using the *Ret^C168* allele. Compared to our RNAi results, we found very similar dendrite defects of C4da neurons

in $Ret^{C168}$ mutant animals (*Figure 2A,B*). These defects were strongly enhanced when we combined $Ret^{C168}$ with a chromosomal deficiency line (*Df(2L)Bsc^{312}*, *Figure 2C*), indicating that $Ret^{C168}$ is indeed a hypomorphic allele. Strikingly, *Ret* mutant C4da neurons displayed incomplete coverage of the receptive field, with dendritic terminals exhibiting patchy distribution, reflecting abnormalities in both shape and growth directionality. In addition, we observed severe isoneuronal dendrite crossing, resulting in extensive overlap of dendritic branches not normally observed in controls (*Figure 2B,C*, arrows). This phenotype is reminiscent of self-avoidance defects described for *Dscam* (*Matthews et al., 2007*; *Soba et al., 2007*; *Hattori et al., 2009*) or tiling mutants like *trc*, *hpo*, and *fry* (*Emoto et al., 2004*, *2006*). To understand the nature of the dendritic crossing phenotype in *Ret* mutant larvae, we performed high resolution confocal live imaging of C4da neurons in vivo. Detailed 3D analysis of dendrite crossing points revealed that virtually all overlapping dendrites were not within the same focal plane, but grew in different planes without directly contacting one another (*Figure 2D, D′,D′′*). Despite the severe patterning and field coverage defects, *Ret* mutant C4da neurons did not show a significant reduction in total dendrite length (*Figure 2E*). They did however display excessive crossings of isoneuronal dendrites outside their normal 2D growth plane (*Figure 2F*) and significantly reduced dendritic field coverage (see Figure 5B,F). Taken together, our results show that *Ret* is

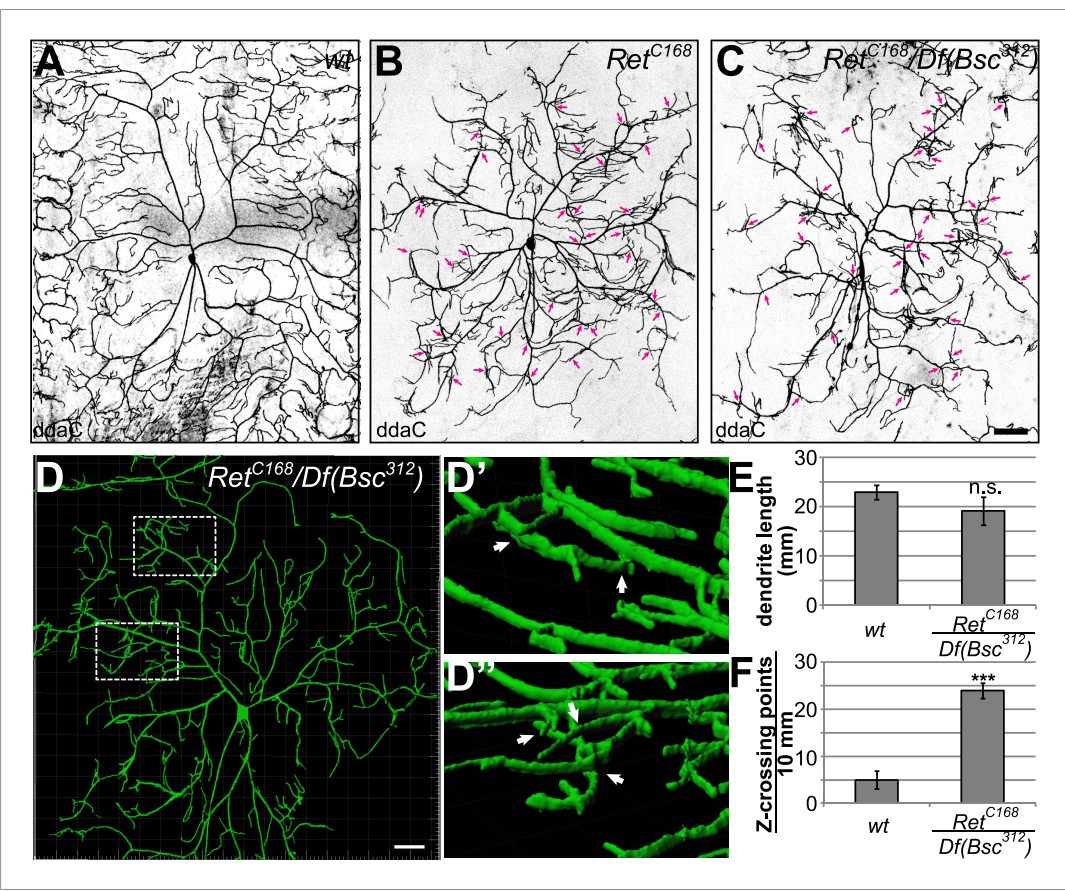

**Figure 2**. Loss of Ret function impairs C4da neuron dendrite patterning and results in out of plane crossing of isoneuronal branches. (**A–C**) Confocal live images of C4da neurons in (**A**) wildtype, (**B**) $Ret^{C168}$ homozygous, and (**C**) $Ret^{C168}/Df(Bsc^{312})$ third instar larvae. Loss of Ret function causes abnormal dendrite patterning featuring reduced receptive field coverage and overlapping dendrites (indicated by arrows). Note that the phenotype is more severe in $Ret^{C168}/Df(Bsc^{312})$ than in $Ret^{C168}$ mutant C4da neurons. Scale bar: 50 μm. (**D**) 3D reconstruction of a *Ret* deficient C4da neuron with magnified 3D views of regions with dendrites crossing in different growth planes (**D′** and **D′′**). Scale bar: 25 μm. (**E**) Total dendrite length of *Ret* mutant C4da neurons was not significantly changed compared to wildtype. (**F**) Out of plane crossing of C4da neuron dendrites in *Ret* mutants was however highly elevated (mean ± SD, n = 8; p < 0.001, Student's two-tailed *t*-test).

required for C4da neuron dendrite patterning and field coverage, and *Ret* loss of function causes out of plane dendritic crossing due to abnormal 3D expansion of dendrites.

## Ret loss of function results in ECM detachment and epithelial enclosure of C4da dendrites

Due to the out of plane crossing observed in *Ret* mutant C4da neurons, we suspected that dendrites were partially detached from the ECM and embedded within the epithelial cell layer. A similar phenotype has recently been described for the integrins *mew* and *mys*, which are required for dendrite-ECM adhesion in da neurons (*Han et al., 2012*; *Kim et al., 2012*). In order to visualize C4da neuron dendrites and the ECM, we took advantage of GFP protein traps for the ECM components *viking* (vkg-GFP) and *trol* (trol-GFP) (*Han et al., 2012*). We performed two-color high resolution confocal imaging of C4da neuron dendrites together with the GFP-labeled ECM in vivo. In wildtype, the majority of dendrites were tightly interacting with the ECM as previously reported (*Figure 3A*, and *Han et al., 2012*). In *Ret* mutant neurons however, large stretches of dendrites, in particular terminal arbors, were detached from the ECM and enclosed by the epidermis (*Figure 3B*). While occasional detachment and out of plane crossing of dendrites could be observed in controls (*Figure 3A′*), the majority of branches were in tight proximity to the ECM (*Figure 3A′′*). *Ret* mutant C4da neurons however displayed frequent detachment and out of plane crossing of dendrites (*Figure 3B′,B′′*) resulting in approx. 11% ECM-detached dendrites within the dorsal field (*Figure 3C*). Our results show that *Ret* is required for robust dendrite-ECM interaction in C4da neurons.

To investigate a possible mechanistic link between dendrite-ECM adhesion defects in *Ret* and *integrin* mutant C4da neurons, we next conducted genetic interaction experiments. While animals heterozygous for *Ret* or *integrin* mutations showed no significant dendrite detachment compared to wildtype, trans-heterozygous combinations of *Ret* and *mys* or *mew* integrin alleles resulted in a significant increase in dendrite crossing defects (not shown) and a substantial loss of dendrite-ECM interaction (*Figure 3D–H*). Similarly to *Ret* mutant C4da neurons, about 10% of dendrites were detached from the ECM in $Ret^{C168}/mew^{M6}$ and $Ret^{C168}/mys^1$ trans-heterozygous animals (*Figure 3I*). These results suggest that *Ret* and integrins act in the same genetic pathway to promote dendrite–matrix interactions.

## Ret is a cell surface protein and interacts with integrins in dendrites

To further assess the link between Ret and integrins, we examined the cellular localization and interaction of Ret with Mys and Mew. We first performed immunohistochemical analysis of larval filet preparations co-expressing Ret together with Mys and Mew in C4da neurons, as integrins endogenous to C4da neurons are difficult to detect due to strong expression in the surrounding epithelial cells (*Han et al., 2012*). To this end, we used a mCherry-tagged Ret transgene which showed a distribution in dendrites similar to endogenous Ret and did not cause obvious phenotypes (*Figure 4—figure supplement 1A*). We found partial colocalization of Ret and Mew or Mys in punctate structures in C4da neuron dendrites (*Figure 4—figure supplements 2, 3*). Colocalized punctae of Ret and integrins could readily be detected in both primary and high order dendritic branches of C4da neurons (*Figure 4—figure supplements 2A′,A′′, 3A′,A′′*). Consistently, transfected S2 cells also displayed colocalization of Ret and Mys, particularly in filopodia-like structures close to the cell surface (*Figure 4—figure supplement 1B*) suggesting that Ret and integrins may form a molecular complex. We therefore performed co-immunoprecipitation experiments of Ret and Mys/Mew in S2 cells. Indeed, Ret showed strong and specific interaction with Mys, and additional co-expression of Mew robustly increased the Ret-integrin interaction indicating that Ret preferentially interacts with the mature integrin complex (*Figure 4—figure supplement 1C*).

To investigate if Ret can also localize to the surface of C4da dendrites and possibly interact with adhesion mediating integrin complexes, we used a tagged Ret transgene carrying an N-terminal pHluorin in addition to the C-terminal mCherry-tag. We performed cell surface immunostainings of Ret using an anti-GFP antibody recognizing the pHluorin-tag together with Mew immunostaining to assess its localization and co-distribution. Ret itself displayed pronounced surface labeling along the entire dendritic tree, strongly indicating that Ret is present at the cell surface of C4da neurons (*Figure 4A,B*). Overall, the pHluorin signal showed surface Ret labeling throughout the dendritic arbor, while the intracellular mCherry-tag revealed additional punctate structures likely reflecting intracellular vesicles (*Figure 4A*). Strikingly, Ret showed significant colocalization with Mew in

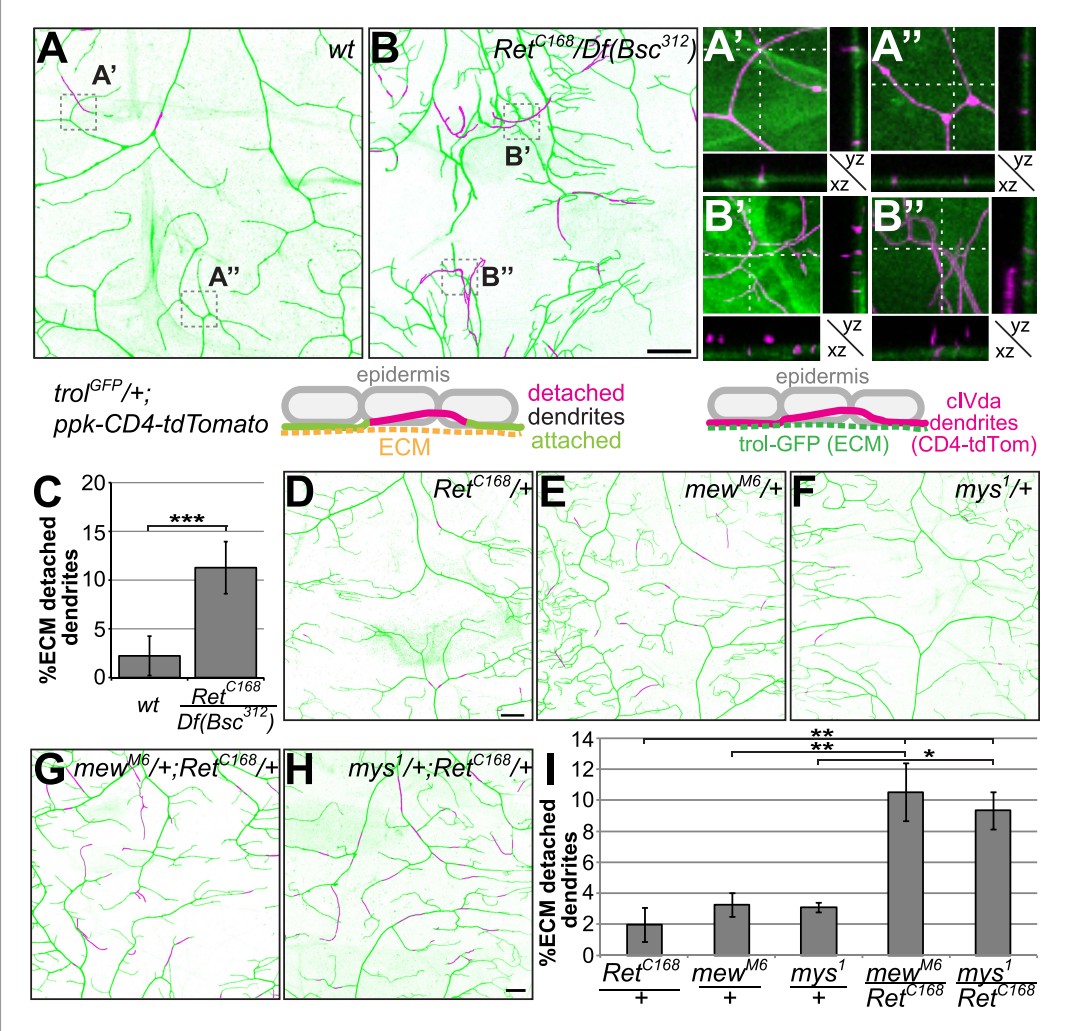

**Figure 3**. Dendrite-ECM detachment due to *Ret* loss of function is genetically linked to integrins. C4da neuron dendrites along the dorsal midline (*ppk-CD4-tdTomato*) and the extracellular matrix (*trol-GFP* or *vkg-GFP*) were co-visualized in third instar larvae. High resolution two-color confocal z-stacks were analyzed for dendrite-ECM interaction and detached dendrite segments are indicated in magenta as indicated schematically. (**A**) In wildtype animals, very few dendrite segments were not in contact with the ECM. Magnified regions of (**A**) and cross-sections illustrate dendrite-ECM proximity with few dendrite segments not contacting the ECM (**A'**, **A''**, ECM in green, dendrites in magenta, see schematic for color code). (**B**) A strong increase of detached dendrites could be observed in *Ret* mutant animals, highlighted in the magnified sections displaying severe displacement of dendrites from the ECM (**B'** and **B''**). Scale bar: 25 μm. (**C**) Quantitative analysis of dendrite-ECM interaction in *wildtype* and *Ret* mutant C4da neurons revealed a strong increase in dendrite detachment in *Ret* deficient animals (mean ± SD, n = 5, p < 0.001, Student's two-tailed *t*-test). (**D–E**) Genetic interaction analysis of dendrite-ECM adhesion in (**D**) *Ret^C168^*, (**E**) *mew^M6^*, (**F**) *mys^1^* heterozygous and (**G**) *mew^M6^/Ret^C168^* (**H**) *mys^1^/Ret^C168^* trans-heterozygous third instar larvae. The combination of either integrin mutant with the *Ret^C168^* allele showed highly increased loss of dendrite-ECM interaction (**G** and **H**) compared to wildtype or the heterozygous alleles alone. Scale bar: 25 μm. (**I**) Quantitative analysis of dendrite detachment for the individual genotypes as indicated. *mew^M6^/Ret^C168^* and *mys^1^/Ret^C168^* trans-heterozygotes showed significantly impaired dendrite-ECM adhesion (mean ± SD, p < 0.05, n = 4, Mann–Whitney U-test).

dendrites (*Figure 4B*). Intensity plots of Ret and Mew signals display a high degree of co-distribution in low and high order dendrites (*Figure 4B',B''*). In particular, many of the high intensity peaks of Mew showed concomitant Ret signal peaks with both tags indicating that Mew and Ret can be indeed colocalized in dendrites and at the surface of C4da neurons. Quantitative colocalization analysis of Ret

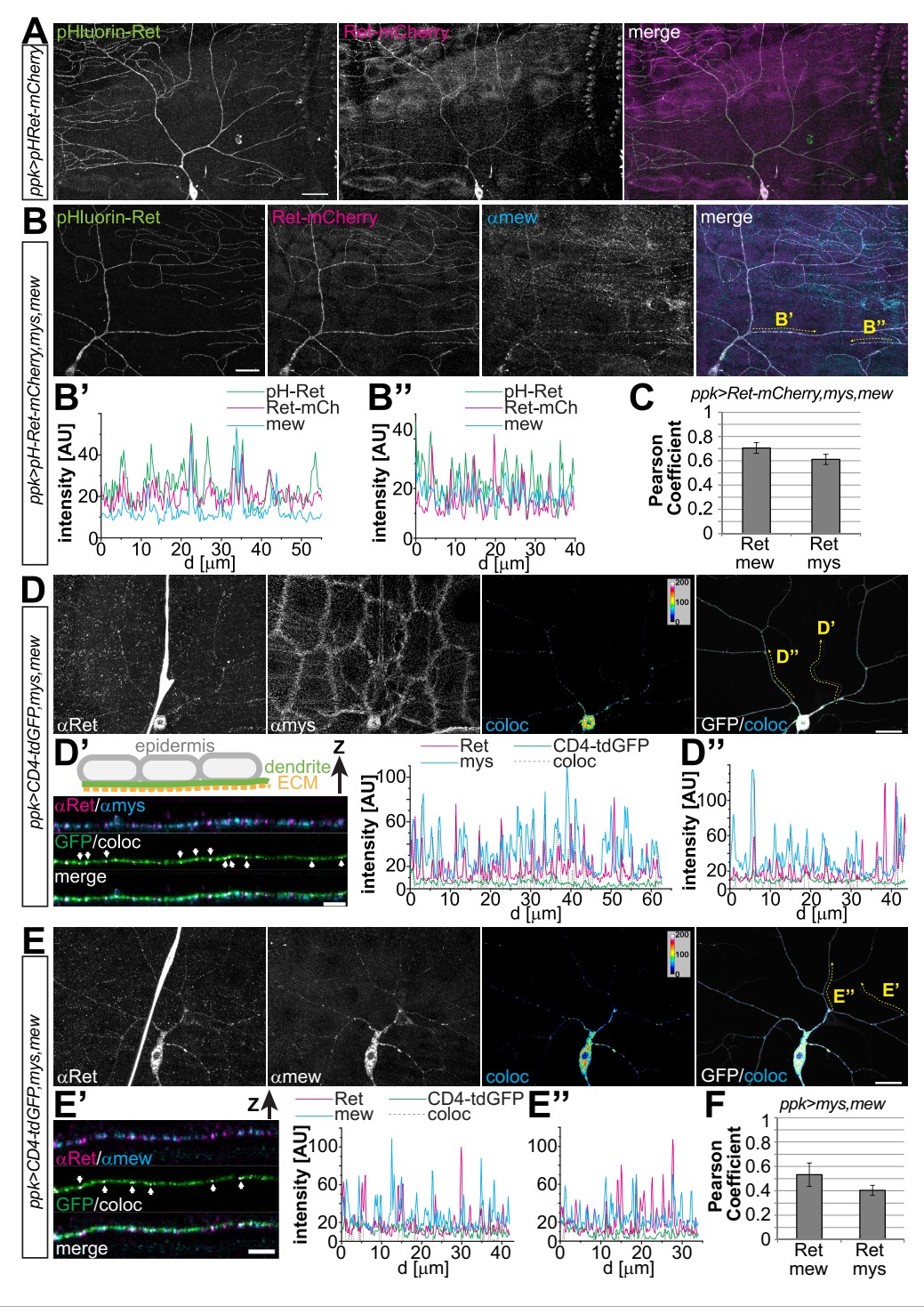

**Figure 4**. Ret localizes to the dendrite surface and co-localizes with integrins. (**A**) Cell surface immunostaining of pHluorin-Ret-mCherry expressed in C4da neurons using *ppk-Gal4*. Surface exposed Ret labeled by pHluorin (surface-stained with anti-GFP antibody) showed even labeling of the entire dendritic tree, while the mCherry signal displayed a more granular distribution of cellular Ret. Scale bar: 30 μm. (**B**) Cell surface immunostaining of pHluorin-Ret-mCherry co-expressing mys and mew in C4da neurons using *ppk-Gal4*. Cell surface and cellular Ret were partially co-distributed with Mew in dendrites, as evident from fluorescence intensity plots along (**B'**) low order and (**B''**) terminal dendrites. Scale bar 20 μm. (**C**) Quantitative colocalization analysis of Ret and integrins coexpressed in C4da neurons (*ppk-Gal4,CD8-GFP > Ret-mCherry,mys,mew*). Pearson coefficients were calculated for C4da neuron
*Figure 4. continued on next page*

*Figure 4. Continued*

soma and dendrite regions showing a positive correlation of Ret and integrin signals (see 'Materials and methods'
for details, mean ± SD, n = 5 per genotype). (**D** and **E**) Colocalization analysis of endogenous Ret and integrins
overexpressed in C4da neurons (*ppk-Gal4,CD4-tdGFP > mys,mew*). Endogenous Ret signal was colocalized with
(**D**) anti-mys or (**E**) anti-mew immunoreactivity in C4da neurons and colocalized pixels visualized in false color
representations (coloc). (**D′** and **E′**) Stretches of terminal dendrites resliced in Z direction showing partial
colocalization of (**D′**) Ret and mys or (**E′**) Ret and mew along the basal dendrite facing the ECM as indicated by
arrows (see schematic drawing). Line intensity plots of the same dendrite portion show signal distribution of
endogenous Ret and integrins together with the colocalized signals (coloc) and the CD4-tdGFP membrane marker.
(**D″** and **E″**) Line intensity plots for a primary dendrite portion (indicated in **D** and **E**). (**F**) Quantitative colocalization
analysis of endogenous Ret and integrins expressed in C4da neurons (*ppk-Gal4,CD4-tdGFP > mys,mew*). Pearson
coefficients calculated for C4da neuron soma and dendrites showing a positive correlation of endogenous Ret and
integrin signals (see 'Materials and methods' for details, mean ± SD, n = 5 per genotype).

The following figure supplements are available for figure 4:

**Figure supplement 1**. Colocalization and biochemical interaction of Ret and integrins in S2 cells.

**Figure supplement 2**. Overexpressed Ret and mys colocalization in C4da neurons.

**Figure supplement 3**. Overexpressed Ret and mew colocalization in C4da neurons.

---

and integrin signals in C4da neuron somata and dendrites revealed a strong positive correlation based
on Pearson coefficient analysis for overexpressed Ret/Mew and Ret/Mys (*Figure 4C*).

Finally, we assessed colocalization of endogenous Ret with overexpressed integrins. As for Ret
overexpression, endogenous Ret displayed a significant degree of co-distribution with Mys and Mew
in C4da neurons (*Figure 4D,E*). Besides pronounced colocalization in the C4da soma, patches of Ret/
Mys and Ret/Mew punctae were detected throughout the dendritic arbor in high (*Figure 4D′,E′*) and
low order branches (*Figure 4D″,E″*). Interestingly, the majority of colocalized signals were found
facing the basal dendritic surface which contacts the ECM (*Figure 4D′,E′*, arrows and *Video 1*)
suggesting that Ret and integrins are potentially interacting at adhesion sites. Overall, quantitative
colocalization analysis of endogenous Ret and integrins showed positively correlated Pearson
coefficients (*Figure 4F*) suggesting that a subset of Ret and integrins is associated in C4da neurons.

Our genetic and immunohistochemical data therefore support the molecular interaction between
Ret and integrins in C4da dendrites. These findings further suggest that Ret and integrins are forming
a common pathway required for normal dendrite growth and ECM adhesion.

## Ret or integrin overexpression in C4da neurons rescues Ret mutant dendrite adhesion defects

Although the Ret-RNAi results and its expression pattern (see *Figure 1*) indicated a cell-autonomous
function of *Ret*, we wanted to test its sufficiency for supporting normal C4da neuron dendrite
morphology. Due to the close proximity of the *Ret* gene to the chromosomal centromere it was not
possible to conduct MARCM analysis (mosaic analysis with a repressible cellular marker [*Lee and Luo,
1999*]). Instead, we performed a rescue experiment by specifically expressing a mCherry-tagged Ret
transgene in C4da neurons in the *Ret* mutant background. Indeed, driving expression of Ret in C4da
neurons was sufficient to completely rescue the dendrite growth and crossing phenotype (*Figure 5B,B′*
and *Figure 5C,C′*) demonstrating that *Ret* is cell-autonomously required for C4da dendrite patterning.

We also asked if overexpression of the α/β-integrin complex is able to rescue dendrite-ECM
interactions in *Ret* mutant animals. To this end, we overexpressed Mys and Mew in *Ret* mutant C4da
neurons and assessed dendritic crossing and field coverage. Although the overexpression of integrins
did not rescue all aspects of the Ret dendritic growth phenotype (*Figure 5D,D′*), it completely
prevented dendritic out of plane crossing indicating restored interaction with the ECM (*Figure 5E*).
However, unlike Ret itself, integrin overexpression in C4da neurons did not rescue coverage defects
of the dendritic field in *Ret* mutant animals (*Figure 5F*). Co-overexpression of Mys and Mew alone did
not cause dendrite abnormalities (data not shown and *Han et al., 2012*) indicating that *Ret* dependent

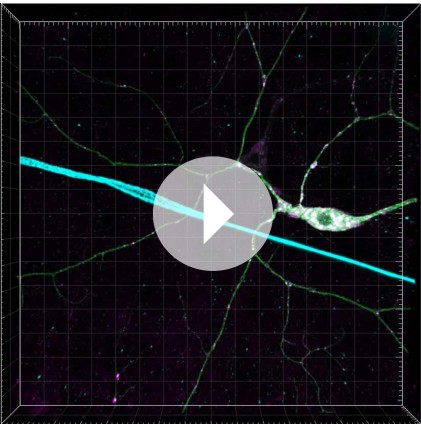

**Video 1.** Colocalization of endogenous Ret and the integrin Mew in C4da neurons. The video shows a 3-dimensional flight view of Ret-Mew colocalization along the C4da dendritic arbor. Shown are CD4-tdGFP labeling the C4da neuron (green), anti-Ret (magenta), and anti-Mew (cyan) together with the colocalized signal (grayscale) between Ret and Mew.

growth involves both integrin dependent and integrin independent pathways. Interestingly, Ret overexpression in the compound eye has been reported to cause photoreceptor degeneration (*Read et al., 2005*). We thus tested if co-expression of integrins could modulate Ret neurotoxicity in photoreceptor neurons (*Figure 5—figure supplement 1*). Integrin over-expression itself did not cause any eye pheno-type, but was able to completely suppress Ret dependent photoreceptor degeneration. These findings show that, similarly to their interactions in C4da neurons, Ret and integrins genetically and functionally interact in the eye.

Our data provide evidence that Ret and the integrins *mys* and *mew* likely operate together in a common pathway to regulate dendrite adhe-sion to the ECM.

## Rac1 loss of function causes loss of dendrite-ECM adhesion and rac1 genetically interacts with Ret and integrins

In order to gain insight into the underlying intracellular signals we investigated candidate pathways common to RTK and integrin signaling. Rho family GTPases, particularly rac1, have been implicated in actin dynamics and dendrite morphogenesis of sensory neurons (*Lee et al., 2003*) and are a common target of both RTK and integrin signaling in many systems (reviewed in *Ivaska and Heino, 2011*). Using loss of function analysis, we tested whether *rac1* is involved in dendrite-ECM adhesion of C4da neurons. Loss of *rac1* function led to strong dendrite-ECM detachment and epithelial enclosure of dendrites highly similar to Ret and integrin mutant phenotypes (*Figure 6A–C*). Overall, in *rac1* mutant C4da neurons, 15% of dendrites in the field were detached from the ECM, an effect that was even more pronounced when the gene dosage of the other two rac-like GTPases, *rac2*, and *mtl* was also reduced (*Figure 6C,D*). In both cases, loss of *rac1* function also led to many non-contacting crossing defects of dendritic branches (*Figure 6B,C* arrows, and *Figure 6E*) indicating that self-avoidance is impaired due to 3D growth of dendrites lacking *rac1* function.

We next tested whether *Ret* and integrins are in the same genetic pathway as *rac1* by analyzing heterozygous allelic combinations. Animals heterozygous for *Ret*, *integrins*, or *rac1* did not show significant changes in C4da neuron dendrite-ECM adhesion compared to wildtype (*Figure 6F–I*). When combining *rac1* alleles with *Ret* or *integrin* mutant alleles however, we observed strong enhancement of dendritic loss of ECM adhesion (*Figure 6J–L*). For both *Ret/rac1* and *integrin/rac1* heteroallelic combinations, 15–20% percent of dendrites lost contact with the ECM indicating strong crosstalk and regulation of Ret and integrins via rac1. These findings strongly suggest that *rac1* functions in the same pathway as *Ret* and integrins in regulating the attachment of dendrites to the ECM.

## Ret mutant C4da neurons display increased dendrite turnover and abnormal F-actin localization

Unlike *Ret*, neither integrin nor *rac1* loss of function resulted in major defects in dendrite growth and coverage of C4da neurons. In addition, integrin overexpression was not able to rescue *Ret* dependent dendrite coverage defects (see *Figure 5*). We reasoned that Ret likely exerts additional functions independent of integrins/rac1. We thus investigated the specific effects of Ret on dendrite growth and turnover. To this end, we performed time lapse analysis during late larval development at 72 hr and 96 hr after egg laying (AEL). This allowed us to monitor dendrite dynamics within a 24 hr period. In wildtype C4da neurons, significant turnover of terminal dendrites could be observed (*Figure 7A*), as reported previously (*Parrish et al., 2007*). Overall, we detected dendrite growth and terminal

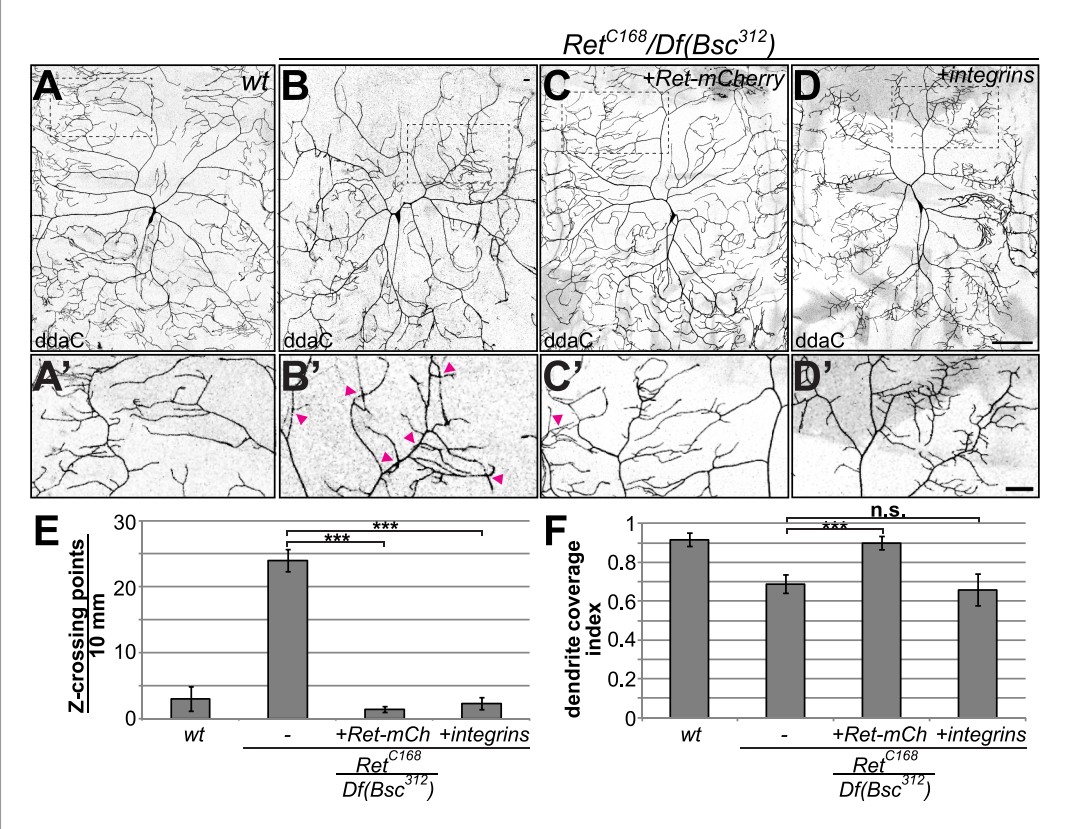

**Figure 5**. Cell-autonomous rescue of *Ret* mutant dendrite defects by C4da-specific expression of Ret or integrins. (**A**–**D**) Maximum projections of C4da neurons visualized with *ppk-Gal4 > CD4-tdGFP* are shown for (**A**) wildtype and (**B**–**D**) *Ret* mutant animals. C4da neuron specific expression of (**C**) *UAS-Ret-mCherry* or (**D**) *UAS-mys/UAS-mew* in *Ret* mutant larvae rescues dendrite crossing defects. However, only re-expression of Ret fully restores dendritic field coverage to wildtype levels. Scale bar 100 μm. (**A'**–**D'**) Magnified view of the indicated dendrite area for the different genotypes. Dendrite crossing points are indicated by arrowheads. Note that Ret-mCherry or integrin expression in *Ret* mutant C4da neurons strongly reduced dendrite crossing events. Scale bar 30 μm. (**E**) Quantitative analysis of out of plane dendrite crossing for the indicated genotypes. Overexpression of Ret or integrins specifically in C4da neurons of *Ret* mutant animals fully rescues dendrite crossing defects (p < 0.001, n = 5) (**F**) Dendrite coverage index (ratio of dendrite field area and segment area (*Parrish et al., 2009*)) is shown for the indicated genotypes. Defects in receptive field coverage of *Ret* mutant C4da neurons are rescued by C4da specific expression of the Ret-mCherry transgene, but not integrin overexpression (mean ± SD, p < 0.001, n = 5, Mann–Whitney U-test).

The following figure supplement is available for figure 5:

**Figure supplement 1**. Integrins suppress Ret overexpression induced adult eye phenotypes.

dynamics required to maintain overall field coverage and tiling during larval growth. In *Ret* mutant animals however, C4da neuron dendrites already showed defects at 48 hr AEL (not shown) and 72 hr AEL, as evident from incomplete coverage and abnormal dendrite patterning at that stage (*Figure 7B*). Increased dendrite crossing was also already apparent. Remarkably, analysis of dendrite turnover between 72 and 96 hr AEL revealed highly increased dynamics of dendritic growth and retraction, with major remodeling of the terminal dendrite arbors. Both growth and retraction of dendrite terminals in *Ret* mutant C4da neurons were increased two to threefold compared to wildtype (*Figure 7C*). Overall, the growth/retraction ratio of dendrites was significantly reduced in *Ret* mutant C4da neurons due to a disproportional increase in retraction compared to growth (*Figure 7D*). Our findings therefore suggest that Ret function is important for stabilizing growing dendrites.

We suspected that the observed increase of dendrite dynamics in *Ret* mutant C4da neurons was linked to the actin cytoskeleton, as integrins, Rac1 as well as Ret are able to directly or indirectly

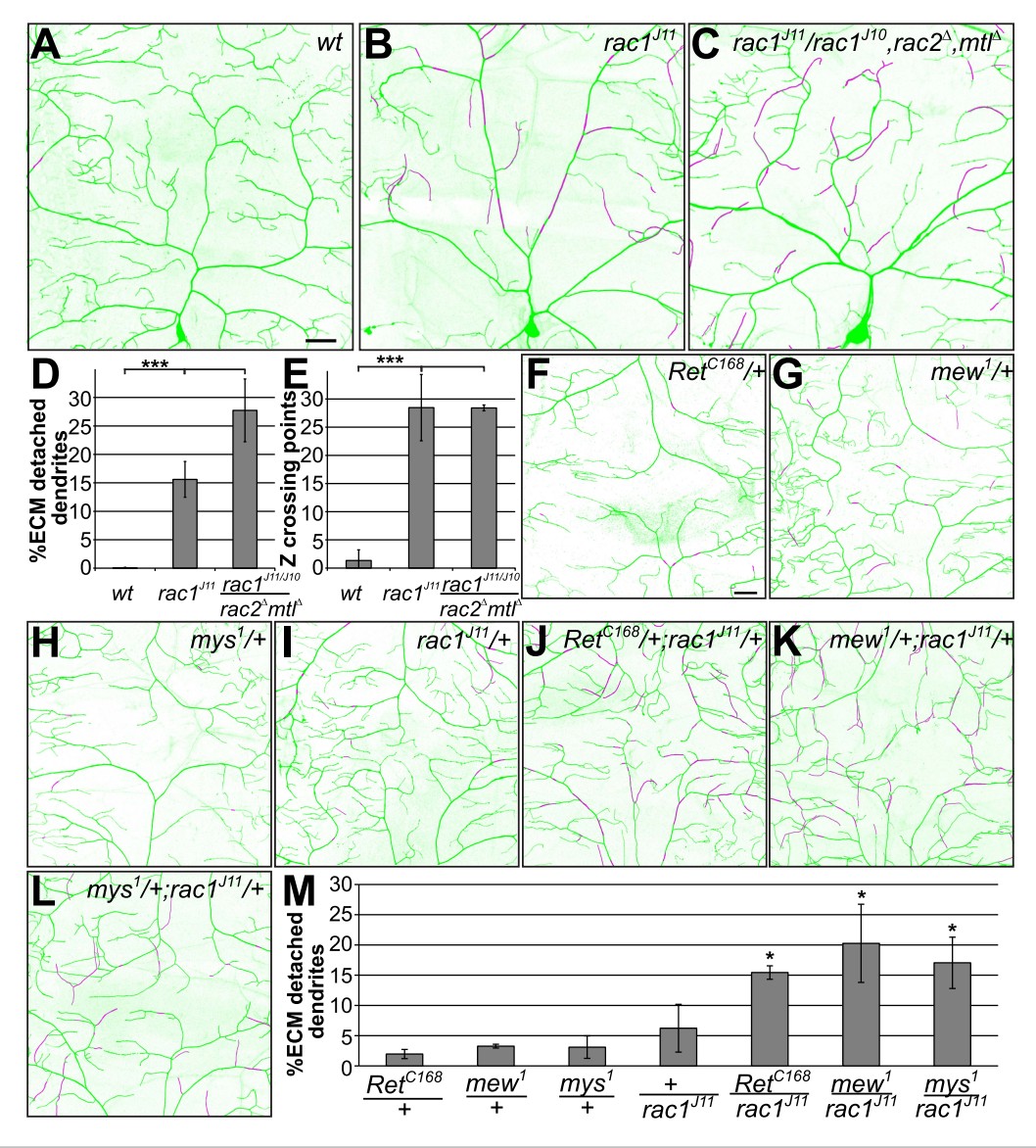

**Figure 6**. *Rac1* loss of function phenocopies and genetically interacts with Ret and integrins. The dorsal field of C4da neurons (*ppk-CD4-Tom*) and the extracellular matrix (*vkg-GFP*) were co-visualized in third instar larvae. High resolution two-color confocal z-stacks were analyzed for dendrite-ECM interaction and enclosed dendrite segments are indicated in magenta. Images of (**A**) Wildtype, (**B**) *rac1^J11* homozygous, and (**C**) *rac1^J11/rac1^J10,rac2^Δ,mtl^Δ* larvae are shown displaying a strong increase of detached dendrites in the *rac1* mutant animals. Scale bar 30 μm. (**D** and **E**) Quantitative analysis of dendrite-ECM interaction and dendrite crossing in *wildtype* and *rac1* mutants shows a significant increase in dendrite detachment (**C**) and Z crossing points (**D**) in *rac1* deficient C4da neurons. (**F–L**) Genetic interaction analysis of dendrite-ECM adhesion in (**F**) *Ret*, (**G**) *mew*, (**H**) *mys*, (**I**) *rac1* heterozygous and (**J**) *rac1/Ret* (**K**) *mew/rac1* (**L**) *mys/rac1* trans-heterozygous third instar larvae. The combination of rac1 mutants with either *Ret* or integrins showed increased loss of dendrite-ECM interaction illustrated by detached portions of the dendritic tree (in magenta). Scale bar 30 μm. (**M**) Quantitative analysis of dendrite detachment from the ECM for the individual genotypes as indicated (mean ± SD, p < 0.05, n = 4 per genotype, Mann–Whitney U-test).

regulate actin (*Fukuda et al., 2002*). To investigate this possibility, we used LifeAct imaging (*Riedl et al., 2008*) to analyze F-actin distribution in dendrites in vivo. Wildtype C4da neurons displayed fairly even distribution of F-actin along major dendrites, while terminal branches typically featured hotspots of increased F-actin levels (*Figure 7E,F*). In contrast, F-actin levels were significantly

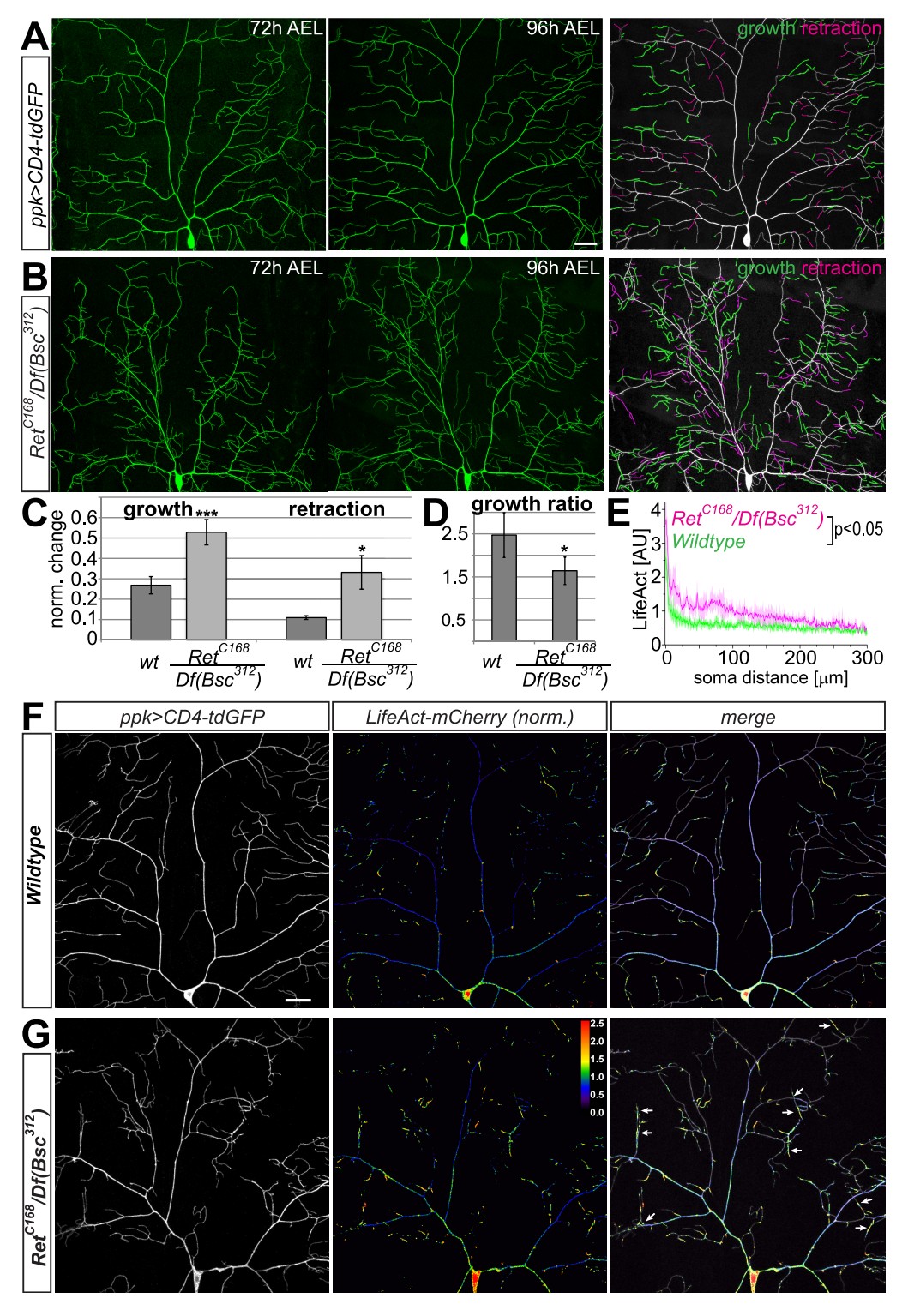

**Figure 7**. Ret mutant C4da neurons display increased dendrite dynamics and aberrant F-actin localization. (**A–B**) The dorsal dendrite field of the same C4da neurons expressing CD4-tdGFP was imaged at 72 hr and 96 hr AEL in (**A**) wildtype and (**B**) Ret mutant animals. Growth and retraction events of dendritic branches were traced and are highlighted in green (growing dendrite) and magenta (retracting dendrite). Scale bar 25 μm. (**C**) Quantitative analysis of dendrite growth and retraction events in wildtype and Ret mutant C4da neurons showing the relative change in dendrite length normalized by the total dendrite length of the dorsal field. Note that growth and retraction of

*Figure 7. Continued*

dendrites in Ret mutant neurons is strongly increased (mean ± SD, p < 0.01 for growth, p < 0.05 for retraction, n = 4). (**D**) Dendrite growth/retraction ratios of *Ret* deficient C4da neurons show a significant decrease compared to wildtype (mean ± SD, p < 0.05, n = 4, Student's *t*-test). (**E**) F-actin levels in primary branches of *Ret* mutant C4da neurons are elevated in proximal regions compared to wildtype. Normalized LifeAct-mCherry signal along primary dendrites was plotted against increasing soma distance showing significantly different intensity profiles (shaded area ± SD, p < 0.05, n = 5). (**F–G**) Dorsal fields of ddaC neurons expressing CD4-tdGFP and LifeAct-mCherry were imaged at 96–100 hr AEL in (**F**) wildtype and (**G**) *Ret* mutant animals. LifeAct levels normalized to GFP are shown as color coded arbitrary intensities. *Ret* mutant C4da neuron dendrites display increased proximal, but reduced distal F-actin levels. Abnormal accumulation of F-actin in aberrant, crossing dendrites was frequently seen in *Ret* mutant but not wildtype neurons (indicated by arrows). Additionally, many terminal branches of *Ret* loss of function neurons have lower levels (below threshold) of F-actin than wildtype. Scale bar 30 μm.

increased and unevenly distributed in proximal regions of primary dendrites of *Ret* mutant neurons (*Figure 7E,G*). The increase in proximal F-actin leveled out at more distal parts of the primary branches, where it reached similar levels in both genotypes (*Figure 7E*). The overall F-actin distribution in major branches of *Ret* mutant C4da neurons was significantly different from wildtype (p < 0.05, n = 5). Strikingly, aberrantly crossing terminal branches had increased levels of F-actin, while other terminals displayed either normal or very low F-actin levels compared to wildtype (*Figure 7G*, arrows). This was also reflected in the centralization and overall uneven distribution of F-actin along dendritic arbors due to the lack of *Ret* function (*Figure 7E*). These data clearly show that Ret is required for the appropriate localization of F-actin in dendrites, which likely contributes to the destabilization of growing branches.

Taken together, our findings show that loss of *Ret* function impairs dendrite stability and increases dynamic growth and retraction of dendritic branches resulting in the failure of complete receptive field coverage. This is likely linked to the observed abnormal distribution of F-actin in the absence of *Ret*, which indicates a lack of appropriate growth signals and abnormal activity of the underlying signaling machinery coordinating actin dynamics.

## Discussion

In this study, we provide evidence that Ret is a regulator of dendrite growth and patterning of C4da neurons. *Ret* is a conserved receptor tyrosine kinase (RTK) expressed in the nervous system of vertebrates (*Pachnis et al., 1993*; *Schuchardt et al., 1994*) and *D. melanogaster* (*Sugaya et al., 1994*; *Hahn and Bishop, 2001*), and has been shown to have a number of important functions in nervous system development and maintenance: it regulates motor neuron axon guidance (*Kramer et al., 2006*), dopaminergic neuron maintenance and regeneration (*Kowsky et al., 2007*; *Kramer et al., 2007*), and mechanoreceptor differentiation and projection to the spinal cord and medulla (*Bourane et al., 2009*; *Luo et al., 2009*). Ret signaling is activated by binding to glial cell line derived neurotrophic factor (GDNF) family ligands and their high affinity co-receptors, the GDNF family receptors (GFRα) (reviewed in *Runeberg-Roos and Saarma, 2007*). Ret also plays an important role in human development and disease as loss of function mutations of Ret lead to Hirschprung's disease displaying colonic aganglionosis due to defective enteric nervous system development (*Amiel et al., 2008*). Conversely, Ret gain of function mutations are causal for autosomal dominant MEN2 (multiple endocrine neoplasia type 2) type medullary thyroid carcinoma (*Lairmore et al., 1993*; *Almeida and Hoff, 2012*).

Prior to this study, Ret has not been implicated in dendrite development. Here, we show that Ret is required specifically for 2D growth of C4da neurons by regulating integrin dependent dendrite-ECM adhesion. Normally, C4da neuron dendrites are virtually always in contact with the ECM and the basal surface of the epithelium lining the larval cuticle, and thus tightly sandwiched between the two compartments (*Yamamoto et al., 2006*; *Han et al., 2012*; *Kim et al., 2012*). In both integrin and *Ret* mutants, dendrite-ECM adhesion is impaired. Ret and integrins can co-localize in dendrites and thus likely form a functional complex that could induce and maintain adhesion of dendrites to the ECM. Since *Ret* loss of function primarily leads to detached terminal dendrite branches, it is tempting to speculate that Ret might be required to recruit integrins to sites of growing dendrites to promote ECM interaction. This is supported by the colocalization of Ret and integrins on the dendrite surface.

Their cooperative interaction could thus ensure proper adhesion of growing branches and, conversely, the fidelity of self-avoidance and tiling.

Our results also highlight the importance of integrating different guidance and adhesion cues to achieve precise neuronal patterning. This has so far only been studied in axon guidance in vivo (*Dudanova and Klein, 2013*). Interestingly, vertebrate *Ret* has been shown to cooperate with Ephrins to ensure high fidelity axon guidance in motor neurons by mediating attractive EphrinA reverse signaling (*Kramer et al., 2006*; *Bonanomi et al., 2012*). Similar mechanisms may conceivably be employed for growing dendrites, which also encounter a multitude of attractive, repulsive, and adhesive cues that have to be properly integrated. Besides pathways acting independently or in a parallel fashion, an emerging view is that receptors exhibit direct crosstalk to integrate incoming signals. So far, only parallel receptor pathways like Dscam and Netrin-Frazzled signaling in class III da neurons (*Matthews and Grueber, 2011*) or Dscam/integrins (*Han et al., 2011*; *Kim et al., 2012*) have been identified co-regulating dendrite morphogenesis. Our data from this study show that the Ret receptor and integrins integrate dendrite adhesion and growth by collaborative interaction of the two cell surface receptors. The molecular and genetic link between *Ret* and integrins suggests that in this case direct receptor crosstalk plays a major role in their function. How exactly these cell surface receptors cooperate and interact remains to be elucidated. Integrins have been shown to display extensive crosstalk with other signaling receptors, including RTKs (*Ivaska and Heino, 2011*). Although integrins are involved in adhesion of virtually all cell types, the underlying signaling and recruitment of integrins to sites of adhesion in vivo is complex and not completely understood. It has been suggested that integrin and growth factor receptor crosstalk can occur by concomitant signaling, collaborative activation, or direct activation of associated signaling pathways (*Ivaska and Heino, 2011*). For example, matrix-bound VEGF can induce complex formation between VEGFR2 and β1-integrin with concomitant targeting of β1-integrin to focal adhesions in endothelial cells (*Chen et al., 2010*). Our findings of biochemical interaction and colocalization of Ret with the α/β-integrins *mys* and *mew* in C4da neuron dendrites argue in favor of direct receptor interaction and subsequent activation of a common signaling pathway.

Integrins and RTKs like Ret do share some of the same intracellular signaling components. These comprise, among others, the MAPK (mitogen-activated protein kinase) pathway, Pi3-Kinase (Pi3K), and Rho family GTPases including Rac1 (*Ivaska and Heino, 2011*). Previous studies provide evidence for Ret-integrin-Rac1 interplay in vitro showing that Ret can enhance integrin mediated adhesion (*Cockburn et al., 2010*) and induce Rac1 dependent lamellipodia formation (*Fukuda et al., 2002*) in cell culture models. In primary chick motor neurons, Rac1 is involved in neurite outgrowth on the integrin substrates laminin and fibronectin (*Kuhn et al., 1998*). Interestingly, Rac1 has previously been shown to regulate dendrite branching in C4da neurons (*Lee et al., 2003*; *Emoto et al., 2004*), however a role in dendrite adhesion in vivo has not been described before. In our study, we show that Rac1 is required for dendrite-ECM adhesion similarly to what has been described for integrins and we genetically link Ret and integrin dependent adhesion with Rac1 function. In *Drosophila*, MAPK, Src and PI3K can be activated by constitutively active Ret overexpression in the compound eye (*Read et al., 2005*; *Dar et al., 2012*). Moreover, novel inhibitors of Ret signaling targeting Raf, Src, and S6-Kinase (S6K) prevent lethality induced by Ret over-activation in a *Drosophila* multiple endocrine neoplasia (MEN2) model (*Dar et al., 2012*). Interestingly, S6K has been shown to be involved in dendrite growth but not tiling in C4da neurons (*Koike-Kumagai et al., 2009*). It remains to be shown if these pathways play a direct role in *Ret* function in dendrite adhesion and growth.

Notwithstanding important commonalities, *Ret* function in C4da neurons cannot be fully explained by crosstalk with integrins and *rac1*. Reduced dendritic field coverage, likely due to the observed increase in dendrite turnover, is only evident in *Ret* but not in integrin or *rac1* mutant C4da neurons. Moreover, increasing integrin expression in a *Ret* mutant background did not rescue dendrite coverage defects, albeit it prevented dendrite crossing. These findings indicate that Ret has additional functions in dendritic branch growth and stability that require as yet unknown extracellular and intracellular mediators. This is also supported by the aberrant F-actin localization in neurons lacking Ret. Here, Ret dependent intracellular effectors are likely important for F-actin assembly to support directed dendrite growth and stabilization, and their localization and activity might be deregulated in the absence of Ret.

*Drosophila Ret* is a highly conserved molecule, its cognate vertebrate ligand GDNF, however is not (*Airaksinen et al., 2006*). In addition, *Drosophila* Ret can neither bind GDNF nor transduce GDNF signaling, although it has been shown to contain a functional tyrosine kinase domain

(*Abrescia et al., 2005*). In mammals, the GFRα co-receptors are essential components of GDNF/Ret signaling (*Runeberg-Roos and Saarma, 2007*). A *Drosophila* GFR-like homolog (dGFRL) has recently been characterized and was found to function and interact with the NCAM homolog FasII (*Kallijärvi et al., 2012*). Therefore, it appears that Ret's functional interaction partners in dendrite development differ significantly from the previously described co-factors in other systems. It is interesting to speculate that a yet undiscovered Ret ligand is involved in Ret mediated dendrite growth and branch stabilization, which might have implications for mammalian Ret function as well: due to its role in the maintenance of dopaminergic neurons and motor axon growth in mouse (*Kramer et al., 2006, 2007*), adhesion related signaling via integrins could well be important during these processes. Moreover, the formation of a dorsal root ganglia derived mechanosensory neurons and their afferent and efferent fiber growth and innervation depends on Ret expression (*Bourane et al., 2009*; *Luo et al., 2009*). It will be interesting to investigate the functional interplay of Ret and integrins in central and peripheral target innervation and neurite maintenance in these systems, given the interdependent function of *Ret* and integrins in sensory dendrite growth as shown in our study.

In summary, we describe a novel role for the Ret receptor in dendrite branch growth and stability in *Drosophila* C4da neurons. This role involves cell-autonomous effects of Ret on ECM adhesion, and F-actin localization in these neurons. Moreover, we have linked dendritic adhesion defects attributable to *Ret* to integrin and *rac1* function featuring a novel and possibly conserved mode of action for Ret in dendrite development.

## Materials and methods

### Fly stocks and transgenic constructs

All fly stocks were maintained at 25°C and 70% rel. humidity on standard cornmeal/molasses food. The following fly stocks were used: $Ret^{C168}$, $Ret^{NP7645}$, $mys^1$, $mew^{M6}$, $rac1^{J11}$, $rac1^{J10}rac2^{\Delta}mtl^{\Delta}$ (*Hakeda-Suzuki et al., 2002*; *Ng et al., 2002*), *ppk-CD4-tdTomato*, *ppk-Gal4*, *UAS-CD4-tdGFP* (*Han et al., 2012*), *trol-GFP*, *vkg-GFP*, *UAS-LifeAct-mCherry* (kindly provided by J Wildonger).

$Ret^{C168}$ is a Piggyback transposon insertion in the 3′UTR of *Ret* and likely a hypomorphic allele leading to reduced Ret protein expression. Df(2L)Bsc$^{312}$ is a deficiency covering the genomic *Ret* locus.

Ret cDNA was amplified from pGMR-Ret (kind gift from Ross Cagan) and cloned into pUAST-attB (*Groth et al., 2004*). In addition, we generated Ret constructs carrying a C-terminal mCherry-tag or an additional pHluorin-tag inserted downstream of the 5′ signal peptide sequence.

*UAS-LifeAct-mCherry*, *UAS-Ret* and *UAS-Ret-mCherry*, *UAS-pHluorin-Ret-mCherry* transgenes were generated by embryo injection into *vasa-ΦC31;attP$^2$* carrying flies according to standard procedures (*Groth et al., 2004*). The *ΦC31*-integrase was outcrossed and transgenes were combined with the appropriate alleles and markers.

### Anti-Ret antibody generation

An antibody against the Ret receptor was generated in guinea pigs by co-injecting the synthetic peptides ETKEVSPGWQAEDAV (peptide 1, corresponding to amino acids 1221–1235 of the intracellular C terminus) and DIHDQATSYDQSEEEM (peptide 2, corresponding to amino acids 1095–1110 of the intracellular C terminus). The resulting antiserum was affinity purified against peptide 1 and used at a 1:1000 dilution. Peptide synthesis, antibody generation, and affinity purification were outsourced to Eurogentec.

### Immunohistochemistry and antibodies

Larval filet preparation and staining was essentially performed as described (*Han et al., 2012*). The following antibodies were used: rabbit anti-phospho-Ret (1:50, Cell Signaling Technology, Danvers, MA), mouse anti-GFP (1:100, Roche Diagnostics, Mannheim, Germany), mouse anti-mys (1:200, Developmental Studies Hybridoma Bank, (DSHB), Iowa City, IO), mouse anti-mew (1:50, DSHB). Secondary DyLight or Alexa conjugated donkey antibodies were from Jackson ImmunoResearch (Westgrove, PA) and were used at 1:400–1:1000.

### Quantitative immunohistochemistry of Ret

Larval filets of wildtype or *Ret* mutant animals carrying a C4da neuron specific marker (*ppk-Gal4 > CD4-tdGFP*) were prepared and immunostained as described above with rabbit anti-phospho-Ret

(1:50, Cell Signaling Technology) or guinea pig anti-Ret (1:1000) antibodies. We quantified the Ret antibody signal of C4da neuron somata in confocal image stacks and calculated signal over background ratios (ΔF/F) by subtracting averaged background signal flanking the C4da soma region (F). Statistical significance for the two genotypes was calculated using a two-tailed t-test.

## Cell surface immunostaining of pHluorin-tagged Ret

For cell surface staining of pHluorin-Ret-mCherry, larval filets were prepared in Ringer solution and incubated with a rabbit anti-GFP antibody (1:50 in Ringer solution, Life Technologies, Carlsbad, CA) for 1 hr on ice. Filets were washed with Ringer solution (3 × 10 min at 4°C) and then fixed with 4% Formaldehyde/PBS for 15 min on ice. Subsequent steps including anti-mew and secondary antibody incubation were carried out in the presence of 0.3% Triton X-100 as above.

## Quantitative Ret and integrin colocalization analysis

Third instar larval filet preparations immunostained for Ret and integrins were analyzed by confocal microscopy with high resolution using a high NA oil objective (Zeiss LSM700, 40×/NA1.3, z step size: 300 nm). Confocal stacks covering the soma and part of the dorsal field of ddaC neurons (xy dimension: 160 × 160 μm) were obtained and deconvoluted using a blind deconvolution algorithm with an adaptive PSF (AutoQuant, BitPlane AG, Zürich, Switzerland). Colocalization analysis was then performed on the deconvolved confocal stacks in 3D using the Imaris Coloc module (BitPlane AG). The GFP reporter signal (ppk-Gal4 > UAS-CD8-GFP or UAS-CD4-tdGFP) was used to create a mask for the C4da neuron soma and dendrite signal to specifically analyze neuronal integrin and Ret signals. Ret and integrin signals were then automatically thresholded and colocalization was calculated (Pearson coefficients, n = 5 per genotype). Costes randomization was performed for all samples to ensure that the calculated colocalization coefficients between Ret and integrins are non-random (n = 100 iterations, p = 1 for all samples).

## Cell culture, immunocytochemistry and biochemistry

S2 cells were grown at 25°C in Schneider's Drosophila medium (Life Technologies) with 10%FBS and Glutamine/Pen/Strep. For experiments, cells were seeded in 6 well plates and transfected at 50% density in an adherent state using Transfectene (Qiagen, Venlo, Netherlands). For S2 cell expression, UAS constructs were co-transfected with pActin-Gal4. The following constructs were used: pUAST-Ret-mCherry, pUAST-mys-3xflag-His, pUAST-mew.

For co-immunoprecipitation experiments, cells were harvested 48 hr after transfection and lysed in 500 μl lysis buffer (50 mM Tris pH7.4, 150 mM NaCl, 1% Triton X-100) for 20 min on ice. After 10 min/4°C/10.000×g centrifugation, supernatants were pre-incubated with mouse IgG Agarose (Sigma–Aldrich, St. Louis, MO) for 30 min at 4°C, and then incubated with anti-flag M2 agarose beads (Sigma–Aldrich) for 4 hr at 4°C. After extensive washing with lysis buffer, the samples were denatured and analyzed on Tris-Acetate gels (Life Technologies) and Western blotting against Ret-mCherry (anti-DsRed, 1:1000, BD Clontech, Mountain Viev, CA) and mys (anti-flag M2, 1:10.000, Sigma).

For immunostaining, S2 cells were allowed to adhere to Concanavalin A (Sigma–Aldrich) coated cover slips for 1 hr and subsequently fixed in 4% Paraformaldehyde/PBS solution for 10 min. After washing with PBS, cells were permeabilized with 0.1% Triton X-100 for 10 min and blocked with 5% donkey serum/PBS. Primary antibodies used were: rabbit anti-DsRed (1:500, BD Clontech), mouse anti-mew (1:100, DSHB), mouse anti-mys (1:100, DSHB). Primary antibodies were applied over night at 4°C in 5% donkey serum/PBS, while secondary donkey antibodies with conjugated DyLight fluorophores (Jackson ImmunoResearch) were subsequently incubated for 1 hr at room temperature. Cells were mounted and imaged by confocal microscopy (Leica SP5 and Olympus FV1000).

## In vivo confocal microscopy and time lapse imaging

C4da neurons of third instar Drosophila larvae (72–100 hr AEL) were imaged alive by confocal microscopy (Leica SP5 or Olympus FV1000), using ppk-CD4-tdTomato or ppk-Gal4, UAS-CD4-tdGFP transgenes (Han et al., 2011). Vkg-GFP or trol-GFP trap lines labeling endogenous vkg or troll proteins with GFP were used to visualize the extracellular matrix. Confocal stacks were taken to image ddaC C4da neuron dendrite fields either with 20× (full ddaC field) or 40× oil objectives (dorsal ddaC

field). For imaging dendrite-ECM interaction, two-color high resolution confocal stacks using a high NA oil objective (Leica or Olympus, 40×, NA 1.3) were taken with a step size of 300 nm.

For time lapse imaging of C4da neurons in wildtype and *Ret* mutant animals, *ppk-Gal4, UAS-CD4-tdGFP* embryos were collected on grape agar plates for 2 hr and allowed to develop at 25°C. Two C4da neurons per animal (abdominal segments a3 and a4) were imaged at 72 hr AEL, and the same neurons were imaged again 24 hr later (96 hr AEL). The imaged larvae were allowed to develop to adulthood to ensure that handling and imaging did not interfere with normal development.

## Analysis of dendrite length, crossing, field coverage, and ECM interaction

Dendrites of C4da neurons were traced with the Imaris Filament Tracer module (BitPlane AG) using deconvoluted confocal stacks. Total dendritic length was calculated from the traces and Z-crossing points were counted and manually confirmed as non-contacting by visual inspection of the z-stack. The high resolution 3D reconstruction was done in Imaris. Dendritic field coverage was calculated essentially as described (*Parrish et al., 2009*) by measuring the area covered by dendrites (polygon formed by connecting all dendritic terminals of the field) divided by the total segment area using Fiji/ImageJ (NIH, Bethesda, MD). Statistical significance was calculated by comparison of all genotypes using a Mann–Whitney test (Origin Pro, Origin Lab, North-hampton, MA).

To quantify net dendrite growth/retraction, confocal Z-projections of the same neuron at 72 and 96 hr AEL were overlayed and adjusted for position and interstitial growth with bUnwarpJ (Fiji, ImageJ). Landmarks marking all major branch points were used to get accurate overlay images. Growing (branch is shorter at 72 hr than 96 hr AEL) and retracting (branch is longer at 72 hr than 96 hr AEL) portions of dendrites were traced and growth/retraction values were normalized to the total length of the unchanged portion of the dendritic tree at 96 hr AEL. Statistical significance was calculated using a Student's two-tailed *t*-test.

Analysis of dendrite-ECM interaction was essentially performed as described (*Han et al., 2012*). Briefly, high resolution confocal stacks were deconvoluted using a blind deconvolution algorithm with an adaptive PSF (AutoQuant, BitPlane AG). Dendrites were traced semi-automatically using the Imaris Filament Tracer module (Bitplane) and thresholded colocalization of the ECM and dendrite signals was performed. Non-contacting and contacting portions of the dendritic tree were traced and verified manually. Percentages of detached dendrites were calculated by the ratios of detached vs total dendrite length. Statistical significance was calculated by comparison of all genotypes using a Mann–Whitney test (Origin Pro).

## Analysis of LifeAct expression and distribution

The dorsal field of C4da ddaC neurons expressing LifeAct-mCherry with *ppk-Gal4 > CD4-tdGFP* was imaged with confocal microscopy at the third instar larval stage (96–100 hr AEL). Acquired stacks were then processed with Fiji/ImageJ (NIH) to normalize the LifeAct signal to GFP expression using a previously described procedure with minor adaptations (*Kardash et al., 2011*). Briefly, we used maximum projections of 2-color stacks and aligned the GFP and LifeAct channels using the TurboReg plugin (ImageJ). The LifeAct signal was thresholded automatically (ImageJ) and normalized to GFP levels using the RatioPlus plugin (ImageJ). For LifeAct quantification along major dendritic arbors, normalized intensity values were obtained by ROI tracing of 2 major branches per sample using the PlotProfile function (ImageJ, n = 5 per genotype). Statistical significance was then calculated by using an Allometric fitting function ($y = a \times x^b$, Origin Pro) for the intensity profiles of wildtype (average $R^2 = 0.74$) and Ret mutant (average $R^2 = 0.79$) samples and fit comparison of the two datasets (p < 0.05, n = 5 per genotype, Origin Pro).

## Acknowledgements

We would like to thank: R Cagan for the *Drosophila* Ret cDNA, J Wildonger for generously sharing the *UAS-LifeAct-mCherry* transgene, and K Duncan for S2 cells; J Parrish, K Duncan, and F Calderon de Anda for comments on the manuscript, the Bloomington, Kyoto and VDRC stock centers for transgenic fly lines; the Developmental Studies Hybridoma Bank (DSHB) and its contributors for the integrin antibodies. LYJ and YNJ are investigators of the Howard Hughes Medical Institute.

## Additional information

### Funding

| Funder | Grant reference | Author |
|---|---|---|
| Howard Hughes Medical Institute (HHMI) | | Lily Yeh Jan, Yuh Nung Jan |
| National Institutes of Health (NIH) | R37NS040929 | Yuh Nung Jan |

The funders had no role in study design, data collection and interpretation, or the decision to submit the work for publication.

### Author contributions

PS, Conception and design, Acquisition of data, Analysis and interpretation of data, Drafting or revising the article; CH, Analysis and interpretation of data, Drafting or revising the article, Contributed unpublished essential data or reagents; YZ, Acquisition of data, Analysis and interpretation of data; DP, IM-A, Drafting or revising the article, Contributed unpublished essential data or reagents; LYJ, YNJ, Conception and design, Drafting or revising the article

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
