## [Decision Letter]

Thank you for sending your work entitled “The Ret receptor regulates sensory neuron dendrite growth and integrin mediated adhesion” for consideration at *eLife*. Your article has been favorably evaluated by K VijayRaghavan (Senior editor) and two reviewers, one of whom is a member of our Board of Reviewing Editors.

The Reviewing editor and the other reviewer discussed their comments before we reached this decision, and the Reviewing editor has assembled the following comments to help you prepare a revised submission.

The manuscript provides evidence that the Ret receptor is involved in controlling three aspects of dendrite growth: self-avoidance, ECM adhesion and field coverage. *Ret* is identified from an RNAi screen designed to search for receptors that regulate dendrite morphogenesis in cIVda neurons in *Drosophila*. *Ret* mutants exhibit increased dendritic crossovers, detachment from the ECM and reduced field coverage. *Ret* mutants also show increased growth dynamics and F-actin in dendrites. Evidence is presented to show that Ret associates with integrin and that *rac1* mediates downstream signalling. Integrin expression rescues the ECM detachment and the self-avoidance phenotypes but not the dendritic field coverage phenotype. The authors conclude that Ret is a novel receptor in the control of dendrite morphogenesis via integrin-dependent and -independent functions.

The reviewers found the finding that Ret is involved in multiple aspects of dendrite growth interesting and novel. The genetic evidence implicating Ret is very compelling, as is the genetic evidence linking Ret with integrin and *rac1* signaling. Some of the expression and colocalization data are less convincing (see specific comments below) and the reviewers called for some further quantitative analysis.

Specific comments: 1) In Figure 1, the Ret mutant image appears to show positive granular labeling yet the text states that the Ret immunostaining is lost in the mutant. Although the signal appears to be in smaller granules than the WT, it is not entirely convincing that there is a significant difference between Figure 1. It would be valuable to include a quantitative analysis of the signal in the WT and mutant.

2) In the subsection headed “Ret is a cell surface protein and interacts with integrins in dendrites”, the authors state that “we could not detect endogenous Ret in higher order dendrites due to limited antibody sensitivity and/or Ret expression levels”, so expression of a transgene, mCherry-Ret, was used to examine the cellular localization of Ret in dendrites and its colocalization with integrin. This is the weakest part of the manuscript. First, in Figure 4, the Ret-mCherry signal is abundant all along the entire dendrite and, therefore, it is bound to show some degree of 'colocalization' with any other molecule that is expressed in the dendrite at the level of resolution and analysis used. Second, the colocalization intensity plots appear to show only one small selected section of a single dendrite. This type of analysis is not very meaningful as it presents n=1 for each condition, is prone to bias (how was each particular section chosen?) and does not provide statistical data over many randomly chosen areas. A more rigorous level of quantitation that provides unbiased analysis over multiple sections of dendrites, giving the Pearson's coefficient values would strengthen the data.

3) The endogenous Ret staining seems fairly robust for the soma and proximal dendrite (Figure 1). The authors could at least confirm Ret-integrin colocalization in the proximal dendrite/soma.

4) Related to the failure to detect endogenous Ret signal in higher order dendrites, can the authors exclude the possibility that Ret is not present here and that the dendritic phenotypes in the *Ret* mutant arise from loss of Ret function in the soma, rather than in the dendrites?

5) How do the authors explain the phenotype? It appears that only late developing, higher order branches are affected by mutations in Ret and its interactions with integrin subunits. Is there a temporal window of action that limits defects to these, or are there other aspects that need investigating?

---

## [Author Response]

*1) In*
Figure 1*, the Ret mutant image appears to show positive granular labeling yet the text states that the Ret immunostaining is lost in the mutant. Although the signal appears to be in smaller granules than the WT, it is not entirely convincing that there is a significant difference between*
Figure 1*. It would be valuable to include a quantitative analysis of the signal in the WT and mutant*.

Following this suggestion, we have first quantified the signal in C4da somata of the phospho-Ret antibody used in our original study. We find that there is indeed a significant difference in anti-phospho-Ret signals between control and *Ret* mutant C4da neurons and have now included the quantitative analysis (see Figure 1—figure supplement 2). Due to the low signal to noise however we have not used this antibody for further studies. Instead, we generated a new Ret specific antibody with higher sensitivity. By using this new antibody, we have added immunostainings of control and *Ret* mutant animals showing specific anti-Ret signal in C4da neurons and quantified the intensities (Figure 1).

*2) In the subsection headed* “*Ret is a cell surface protein and interacts with integrins in dendrites*”*, the authors state that* “*we could not detect endogenous Ret in higher order dendrites due to limited antibody sensitivity and/or Ret expression levels*”*, so expression of a transgene, mCherry-Ret, was used to examine the cellular localization of Ret in dendrites and its colocalization with integrin. This is the weakest part of the manuscript. First, in*
Figure 4*, the Ret-mCherry signal is abundant all along the entire dendrite and, therefore, it is bound to show some degree of 'colocalization' with any other molecule that is expressed in the dendrite at the level of resolution and analysis used*.

We agree that the original conclusions using Ret-mCherry transgene were limited. Using our new Ret antibody we could show that endogenous Ret is indeed present throughout the dendritic arbor of C4da neurons, even in high order branches (Figure 1). Thus endogenous Ret and overexpressed Ret-mCherry have a similar distribution within dendrites (see Figure 1 and Figure 4). Although, with the very broad distribution of overexpressed Ret, there is bound to be some degree of colocalization between Ret and any other protein in dendrites, the resolution of our images is not the limiting factor (156x156x300 nm voxels). We did address this issue by extensively quantifying the degree of colocalization between endogenous or overexpressed Ret and integrins in C4da showing a robust positive correlation between Ret and integrin localization (please see responses below).

*Second, the colocalization intensity plots appear to show only one small selected section of a single dendrite. This type of analysis is not very meaningful as it presents n=1 for each condition, is prone to bias (how was each particular section chosen?) and does not provide statistical data over many randomly chosen areas. A more rigorous level of quantitation that provides unbiased analysis over multiple sections of dendrites, giving the Pearson's coefficient values would strengthen the data*.

We appreciate this helpful comment and suggestion. Our intention with the intensity plots was to illustrate the signal distribution of Ret and integrins in low and high order branches. We agree that this is not quantitatively meaningful and have thus included extensive colocalization analysis for the original data co-expressing Ret-mCherry and integrins in C4da neurons, (see Figure 4 and Figure 4—figure supplement 2 and Figure 4—figure supplement 3). We analyzed colocalization in 3D in the soma region and all dendrites of C4da neurons in high resolution confocal image stacks (160x160 μm fields) that were deconvolved and controlled for signal intensities and bleed-through (please see new section in Material and methods for details). We now provide Pearson coefficient analysis of entire C4da neuron soma and dorsal dendrite fields. Our analysis revealed that there is indeed a positive correlation of Ret and integrin signals in C4da neurons based on Pearson coefficients suggesting that a meaningful subset of Ret and integrins are colocalized (Figure 4).

*3) The endogenous Ret staining seems fairly robust for the soma and proximal dendrite (*Figure 1*). The authors could at least confirm Ret-integrin colocalization in the proximal dendrite/soma*.

We agree that endogenous colocalization is a more significant measure for protein proximity and interaction. We thus used our newly developed Ret antibody for colocalization analyses. Unfortunately, we could not circumvent integrin overexpression in C4da neurons, as the epithelial expression of *mys* and *mew* is significantly higher than in C4da neurons thus preventing us to visualize it directly, even in the soma region. Only upon integrin overexpression we could unambiguously discern the neuronal signal and used optimized conditions to analyze colocalization with endogenous Ret. The new data and analysis is presented in Figure 4. Similar to the Ret/integrin co-overexpression results, we did find significant colocalization of endogenous Ret and integrins in the C4da soma and in dendrites. We show representative results using colocalization channels, line scan Z stacks and line intensity plots for *Ret/mys* (Figure 4) and *Ret/mew* (Figure 4). In addition, quantitative assessment of Pearson coefficients (as described in #2) showed a consistently positive correlation of Ret and integrin colocalization in C4da neuron somata and dendrites (Figure 4). In addition, we found that endogenous Ret localizes preferentially to the basal side of dendrites which is in contact with the extracellular matrix (ECM, see Figure 1). Coincidentally, we also detect most of the colocalized Ret-integrin signal facing the ECM (see Figure 4 and Video 1).

Taken together with our additional genetic and biochemical evidence, we believe that our data supports the view that a subset of Ret and integrins interacts in dendrites to regulate dendrite-ECM adhesion. However, we would like to stress that we do not think that Ret and integrins are constitutive interaction partners, but rather transiently interacting at specific sites to promote adhesion upon signaling. Although this is speculative there is precedence in the literature for transient integrin-RTK interaction and recruitment to adhesion sites (see paragraph 4 of our Discussion).

*4) Related to the failure to detect endogenous Ret signal in higher order dendrites, can the authors exclude the possibility that Ret is not present here and that the dendritic phenotypes in the* Ret *mutant arise from loss of Ret function in the soma, rather than in the dendrites*?

The reviewers raised an interesting point here. We believe that our new data strongly argues in favor of Ret having a function in dendrites. Firstly, our endogenous Ret staining now shows specific presence of Ret in all dendrites of C4da neurons. Secondly, our *Ret* mutant still displayed some somatic Ret expression but dendritic Ret levels dropped below the detection limit. Nonetheless, striking dendrite phenotypes are seen in this background. Although we cannot and would not like to exclude that somatic Ret has an important function, we believe that these data argue in favor of Ret being required in dendrites to regulate adhesion and dendrite dynamics.

*5) How do the authors explain the phenotype? It appears that only late developing, higher order branches are affected by mutations in Ret and its interactions with integrin subunits. Is there a temporal window of action that limits defects to these, or are there other aspects that need investigating*?

This is an interesting point. Indeed, the loss of Ret seems to mostly affect later developing terminal dendrites. However, dendrite crossing and growth defects are already evident at earlier stages (at least at 48h AEL). We observed numerous stretches of lower order dendrites that were detached from the ECM at the third instar stage. Whether Ret is actually required for maintaining dendrite-ECM attachment or whether these dendrites were already detached when they were originally growing remains to be investigated. Although technically challenging, this would be feasible and interesting to do in future studies.